# Branch & Learn for Recursively and Iteratively Solvable Problems in Predict+Optimize

**Xinyi Hu[1], Jasper C.H. Lee[2], Jimmy H.M. Lee[1], Allen Z. Zhong[1]**
[1]Department of Computer Science and Engineering
The Chinese University of Hong Kong, Shatin, N.T., Hong Kong
[2]Department of Computer Sciences & Institute for Foundations of Data Science
University of Wisconsin–Madison, WI, USA
{xyhu,jlee,zwzhong}@cse.cuhk.edu.hk, jasper.lee@wisc.edu

## Abstract

This paper proposes *Branch & Learn*, a framework for Predict+Optimize to tackle optimization problems containing parameters that are unknown at the time of solving. Given an optimization problem solvable by a recursive algorithm satisfying simple conditions, we generalize previous work substantially by showing how a corresponding learning algorithm can be constructed directly and methodically from the recursive algorithm. Our framework applies also to iterative algorithms by viewing them as a degenerate form of recursion. Extensive experimentation shows better performance for our proposal over classical and state of the art approaches.

## 1 Introduction

In the intersection of machine learning and constrained optimization, the Predict+Optimize framework tackles optimization problems with parameters that are unknown at solving time. Such uncertainty is common in daily life and industry. For example, retailers need to pick items to restock for maximizing profit, yet consumer demand is a-priori unknown.

The task is to i) predict the unknown parameters, then ii) solve the optimization problem using the predicted parameters, such that the resulting solutions are good even under true parameters. Traditionally, the parameter prediction uses standard machine learning techniques, with error measures independent of the optimization problem. Thus, the predicted parameters may in fact lead to a low-quality solution for the (true) optimization problem despite being "high-quality" for the error metric. The Predict+Optimize framework uses the more effective *regret function* [3, 6, 9] as the error metric, capturing the difference in objective (computed under the true parameters) between the estimated and true optimal solutions. However, the regret function is usually not (sub-)differentiable, and gradient-based methods do not apply.

Prior works have focused on the regime where the optimization problems contain unknown objective and known constraints, and proposed ways to overcome the non-differentiability of the regret. They can be roughly divided into two approaches: *approximation* and *exact*. The former tries to compute the (approximate) gradients of (approximations of) the regret function. Elmachtoub *et al.* [6] propose a differentiable surrogate function for the regret function, while Wilder *et al.* [17] relax the integral objective in constrained optimization and solve a regularized quadratic programming problem. Mandy and Guns [12] focus on mixed integer linear programs and propose an interior point based approach. While novel, approximation approaches are not always reliable. *Exact* approaches exploit the structure of optimization problems to train models without computing gradients. Demirović *et al.* [4] investigate problems with the ranking property and propose a large neighborhood search method to learn a linear prediction function. They [5] further extend the method to enable Predict+Optimize for problems amenable to tabular dynamic programming (DP).

36th Conference on Neural Information Processing Systems (NeurIPS 2022).

We propose a novel *exact* method for problems solvable with a recursive algorithm (under some restrictions), significantly generalizing the work of Demirović *et al.* [5]. By viewing iteration as a special case of recursion with a single branch, our framework applies also to iterative algorithms. Recall that DP is a special case of recursion, where the recursion structure has overlapping sub-problems, enabling the use of memoization to avoid recomputation. In particular, tabular DP is implemented as an iterative algorithm, computing the table row by row. While tabular DP is a widely applicable technique, it is not universal in that many naturally recursively solvable problems are not known to have a DP algorithm. For example, many modern combinatorial search algorithms are still fundamentally based on exhaustive search. Our work thus subsumes the method by Demirović *et al.* [5] and extends the Predict+Optimize framework to a much wider class of optimization problems. Experiments on 3 benchmarks with artificial and real-life data against 9 other learning approaches confirm the superior solution quality and scalability of our method.

The concurrent work by Guler *et al.* [9], just accepted to AAAI22, proposes a divide and conquer algorithm, extending the work of Demirović *et al.* [5] by considering optimization problems whose objective function is a bilinear function in the decision variables and the unknown parameters. While both Branch & Learn and [9] are exact methods for Predict+Optimize, there are problems amenable to Branch & Learn but not [9], and vice versa. The techniques used are also very different between the two works. Section 6 compares the two pieces of work in more detail.

## 2 Background

Without loss of generality, an *optimization problem* is to find $x^* = \arg\min_x obj(x)$ s.t. $C(x)$, where $x \in \mathbb{R}^d$ is a vector of decision variables, $obj : \mathbb{R}^d \to \mathbb{R}$ is a function mapping $x$ to an *objective value* to be minimized, and $C$ is a set of constraints over $x$. Thus, $x^*$ is an *optimal solution* and $obj(x^*)$ is the *optimal value*. In this (and prior) work, we focus on cases where only the objective contains uncertainty. A *parameterized optimization problem (Para-OP)* $P(\theta)$ thus extends the problem $P$ as:

$$x^*(\theta) = \arg\min_x obj(x, \theta) \text{ s.t. } C(x)$$

where $\theta \in \mathbb{R}^t$ is a vector of parameters. The objective now depends also on $\theta$. When the parameters are known, a Para-OP is just an optimization problem.

**Example 1.** *Consider a project funding problem to maximize $\sum_{i=1}^{4} p_i \cdot x_i$ subject to the budget constraint $2x_1 + 2x_2 + x_3 + x_4 \leq 3$, where $p$ is an array representing the profits of projects. The problem is an instance of 0-1 knapsack. However, $p$ is usually unknown at decision time, with only some features related to it given, such as proposal scores and the reputation of each applicant.*

In *Predict+Optimize* [5], the true parameters $\theta \in \mathbb{R}^t$ for a Para-OP are unknown at solving time, and *estimated parameters* $\hat{\theta}$ are used instead. Suppose each parameter is estimated by $m$ features. The estimation will rely on a machine learning model trained over $n$ observations of a training data set $\{(A^1, \theta^1), \ldots, (A^n, \theta^n)\}$, where $A^i \in \mathbb{R}^{t \times m}$ is a *feature matrix* for $\theta^i$, so as to yield a *prediction function* $f : \mathbb{R}^{t \times m} \to \mathbb{R}^t$ for estimating parameters $\hat{\theta} = f(A)$.

The quality of the estimated parameters $\hat{\theta}$ is measured by the *regret function*, which is the objective difference between the *true optimal solution* $x^*(\theta)$ and the *estimated optimal solution* $x^*(\hat{\theta})$ under the true parameters $\theta$. Formally, we define the regret function $Regret(\hat{\theta}, \theta) : \mathbb{R}^t \times \mathbb{R}^t \to \mathbb{R}_{\geq 0}$ to be:

$$Regret(\hat{\theta}, \theta) = obj(x^*(\hat{\theta}), \theta) - obj(x^*(\theta), \theta)$$

where $obj(x^*(\hat{\theta}), \theta)$ is the *estimated optimal value* and $obj(x^*(\theta), \theta)$ is the *true optimal value*. Following the empirical risk minimization principle, Elmachtoub *et al.* [7] choose the prediction function from the set of models $\mathcal{F}$ to attain the smallest average regret over the training data:

$$f^* = \arg\min_{f \in \mathcal{F}} \frac{1}{n} \sum_{i=1}^{n} Regret(f(A^i), \theta^i) \tag{1}$$

For discrete optimization problems, the regret is not (sub) differentiable. Hence, traditional (sub-)gradient-based training algorithms are not applicable.

---

**Algorithm 1:** Coordinate Descent

---

**Input:** A Para-OP $P(\theta)$ and a training data set $\{(A^1, \theta^1), \ldots, (A^n, \theta^n)\}$
**Output:** a coefficient vector $\alpha \in \mathbb{R}^m$

1 Initialize $\alpha$ arbitrarily and $k \leftarrow 0$;
2 **while** *not converged* $\wedge$ *resources remain* **do**
3     $k \leftarrow (k \mod m) + 1$;
4     Initialize $L$ to be the zero constant function;
5     **for** $i \in [1, 2, \ldots, n]$ **do**
6         $(P^i_\gamma, I_0) \leftarrow \texttt{Construct}(P(\theta), k, A^i)$ ;
7         $E^i(\gamma) \leftarrow \texttt{Convert}(P^i_\gamma, I_0)$;
8         $L^i(\gamma) \leftarrow \texttt{Evaluate}(\mathbb{I}(E^i), \theta^i, I_0)$;
9         $L(\gamma) \leftarrow L(\gamma) + L^i(\gamma)$;
10     $\alpha_k \leftarrow \arg\min_{\gamma \in \mathbb{R}} L(\gamma)$;

11 return $\alpha$;

---

Demirović *et al.* [5] study the class $\mathcal{F}$ of linear prediction functions and propose to represent the solution structure of a Para-OP using (continuous) piecewise linear functions. A *piecewise linear function* $h$ is a real-valued function defined on a finite set of (closed) intervals $\mathbb{I}(h)$ partitioning $\mathbb{R}$. Each interval $I \in \mathbb{I}(h)$ is associated with a linear function $h[I]$ of the form $h[I](r) = a_I r + b_I$, and the value of $h(r)$ for a real number $r \in \mathbb{R}$ is given by $h[I](r)$ where $r \in I$. An algebra can be canonically defined on piecewise linear functions [15]. For piecewise linear functions $h$ and $g$, we define pointwise addition as $(h + g)(r) = h(r) + g(r)$ for all $r \in \mathbb{R}$. Pointwise subtraction, max/min and scalar products are similarly defined. All five operations can be computed efficiently by iterating over intervals of the operands [5].

In the rest of the paper, we assume that the prediction function $f$ is a *linear mapping of the form* $f(A) = A\alpha$ for some $m$-dimensional vector of *coefficients* $\alpha \in \mathbb{R}^m$.

To (approximately) solve problem (1), Demirović *et al.* [5] proposes to update the coefficients $\alpha$ of $f$ iteratively via coordinate descent (Algorithm 1), in which $\alpha$ are initialized arbitrarily and updated in a round-robin fashion. Each iteration (lines 3-10) contains three functions. $\texttt{Construct}$ constructs a Para-OP as a function of the *free coefficient*, fixing the other coefficients in $\alpha$, with an initial domain $I_0$. $\texttt{Convert}$ returns a piecewise linear function of the free coefficient from the Para-OP, and each interval of the function corresponds to an estimated optimal solution. $\texttt{Evaluate}$ takes the returned function and the true parameters as inputs, computes the estimated optimal value, and obtains the regret as a piecewise constant function of the free coefficient.

As Algorithm 1 shows, in each iteration (lines 3-10), an coefficient $\alpha_k$ is updated. Iterating over index $k \in \{1, \ldots, m\}$, we replace $\alpha_k$ in $\alpha$ with a variable $\gamma \in \mathbb{R}$ by constructing $\alpha + (\gamma - \alpha_k)e_k$, where $e_k$ is a unit vector for coordinate $k$. In lines 5-10, we wish to update $\alpha_k$ as:

$$\alpha_k \leftarrow \arg\min_{\gamma \in \mathbb{R}} \sum_{i=1}^n Regret(A^i e_k \gamma + A^i(\alpha - \alpha_k e_k), \theta^i)$$

Let us describe lines 6-8 in more detail. For notational convenience, let $a^i = A^i e_k \in \mathbb{R}^m$ and $b^i = A^i(\alpha - \alpha_k e_k) \in \mathbb{R}^m$, which are vectors independent of the free variable $\gamma$. $\texttt{Construct}$ synthesizes the parameterized problem

$$P^i_\gamma \equiv x^*(a^i\gamma + b^i) = \arg\min_x obj(x, a^i\gamma + b^i) \text{ s.t. } C(x)$$

Sometimes, the parameterized problem can also have an initial domain $I_0 \neq \mathbb{R}$ for $\gamma$. For instance, we may restrict the estimated profits to be non-negative in Example 1. $\texttt{Convert}$ takes $P^i_\gamma$ to create a function $E^i$ mapping $\gamma$ to the estimated objective $E^i(\gamma) = obj(x^*(a^i\gamma + b^i), a^i\gamma + b^i)$. Associated with each interval $I \in \mathbb{I}(E^i(\gamma))$, a linear function maps $\gamma$ to the objective computed with the estimated parameters $a^i\gamma + b^i$, and the estimated optimal solution $x^*(a^i\gamma + b^i)$ remains the same in each interval $I$ [4, 5]. From this, $\texttt{Evaluate}$ computes the regret $L^i$ for each interval $I$, i.e.

$$L^i[I] = Regret(a^i\chi(I) + b^i, \theta^i) = obj(x^*(a^i\chi(I) + b^i), \theta^i) - obj(x^*(\theta^i), \theta^i)$$

---

**Algorithm 2:** Generic Recursive Solving and Recursive Learning

| | |
|---|---|
| **1 Function** $ReSolve(P, \theta)$**:** | **10 Function** $ReLearn(P_\gamma, I_0)$**:** |
| **2**    **if** $isBaseCase(P)$ **then** | **11**    **if** $isBaseCase(P_\gamma, I_0)$ **then** |
| **3**       $(x^*(\theta), C^*(\theta)) \leftarrow \texttt{BaseCase}(P, \theta);$ | **12**       $R[I_0] \leftarrow \texttt{BaseCaseL}(P_\gamma);$ |
| **4**    **else** | **13**    **else** |
| **5**       $T \leftarrow \texttt{Extract}(P, \theta);$ | **14**       $T_\gamma \leftarrow \texttt{ExtractL}(P_\gamma, I_0);$ |
| **6**       $PL \leftarrow \texttt{Branch}(P, \theta, T);$ | **15**       **for** *each interval* $I \in \mathbb{I}(T_\gamma)$ **do** |
| **7**       $RL \leftarrow \texttt{Map}(ReSolve, PL);$ | **16**          $PL \leftarrow \texttt{BranchL}(P_\gamma, T_\gamma[I]);$ |
| **8**       $(x^*(\theta), C^*(\theta)) \leftarrow \texttt{Reduce}(\oplus, RL);$ | **17**          $RL \leftarrow \texttt{Map}(ReLearn, PL);$ |
| **9**    **return** $R$; | **18**          $R[I] \leftarrow \texttt{Reduce}(\boxplus, RL);$ |
| | **19**    **return** $R$; |

---

where $\chi$ is a function arbitrarily selecting a value from an interval $I$. The resulting function $L^i$ is piecewise constant and represents the regret for different values of the free coefficient. If $\gamma \notin I_0$, the value of $L^i$ is set to a sufficiently large constant to indicate impracticability of the estimation $\hat{\theta} = a^i \gamma + b^i$. Line 9 sums all $L^i(\gamma)$ into a piecewise constant total regret $L(\gamma)$ across all training examples, and line 10 minimizes $L(\gamma)$ by simply iterating over each interval of $L(\gamma)$.

While coordinate descent is a standard technique, the key contribution by Demirović *et al.* [5] is to show how to build the `Convert` function for a tabular DP algorithm, by computing a modified DP table with essentially the same structure. In this paper, we *significantly generalize* the method by constructing `Convert` for recursive and iterative algorithms, opening the Predict+Optimize framework to *a much wider class* of problems than just those solvable using DP algorithms. A major obstacle to the generalization is that different prediction coefficients $\gamma$ can lead to wildly different executions of the recursion, all of which need to be explored in order to optimize for $\gamma$ minimizing the empirical regret. At a high level, we overcome this challenge by presenting a standard template for recursive algorithms (ReSolve in Algorithm 2), which we show can be cleanly adapted (ReLearn) to algorithmically enumerate all the possible recursion executions based on the free coefficient $\gamma$.

## 3 Recursively Solvable Problems

This section describes the general form (ReSolve in Algorithm 2) of recursive algorithms considered, which in particular captures also all DP algorithms without memoization applied. This template uses the following two higher-order functions:

**Definition 1** (Map). *Suppose $D, S$ are two sets and $f : D \rightarrow S$ is a function.* $\texttt{Map}(f, [d_1, \ldots, d_l])$ *returns a list $[f(d_1), \ldots, f(d_l)]$ where $d_1, \ldots, d_l \in D$ and $f(d_i) \in S$.*

**Definition 2** (Reduce). *Suppose $S$ is a set and $\oplus : S \times S \rightarrow S$ is a commutative and associative operation.* $\texttt{Reduce}(\oplus, [s_1, s_2, \ldots, s_l])$ *returns $s_1 \oplus s_2 \oplus \cdots \oplus s_l$.*

The ReSolve function takes a recursively solvable Para-OP $P$ and known parameters $\theta$, and returns the optimal value and decisions $C^*$ and $x^*$ for $P(\theta)$. It has several key components: i) isBaseCase checks if $P(\theta)$ is a base case, and if so, BaseCase returns the result of $P(\theta)$. ii) Extract, using the current problem $P$ and current parameters $\theta$, computes some information $T$ which will determine the list of subproblems. iii) Branch, using only $T$, creates and returns a list $PL$ of subproblems, and computes the corresponding parameters from both $T$ and the current parameters $\theta$. iv) Each subproblem in $PL$ is solved recursively, via Map mapping ReSolve to $PL$. After the recursive calls, the partial results in $RL$ are aggregated by a binary operation $\oplus$ via Reduce. As mentioned, the ReSolve template also captures iterative algorithms as recursion with a single branch.

We also restrict attention to algorithms satisfying: i) the only arithmetic operations involving the unknown parameters are $+, -, \max, \min$ and multiplication with known constants, ii) there are no conditionals within Branch, and iii) isBaseCase is independent of the parameters $\theta$.

**Example 2.** *`ReSolve_KS` in Algorithm 3 is an instantiation of `ReSolve` for solving the project funding (0-1 knapsack) problem in Example 1. `ReSolve_KS` takes input $(p, c, n, W, S)$ where $n$ is the number of remaining projects to consider, $W$ is the available funding, and $S$ are the*

---
**Algorithm 3:** Recursive Solving and Recursive Learning for 0-1 Knapsack Problem
---

**1 Function** $ReSolve\_KS(p, c, n, W, S)$:
  **2**   **if** $n = 0$ *or* $W \le 0$ **then**
  **3**     $R \leftarrow \mathbb{1}(W \ge 0) \cdot \sum_{i \in S} p[i]$;
  **4**   **else**
  **5**     $[P_1, P_2] \leftarrow$ Branch_KS$(p, c, n, W, S)$;
  **6**     $RL \leftarrow$ Map(Resolve_KS, $PL$);
  **7**     $R \leftarrow$ Reduce(max, $RL$);
  **8**   **return** $R$;

**9 Function** $ReLearn\_KS(p_\gamma, c, n, W, S, I)$:
  **10**   **if** $n = 0$ *or* $W \le 0$ **then**
  **11**     $R[I] \leftarrow \sum_{i \in S} p_\gamma[i]$;
  **12**   **else**
  **13**     $[P_1, P_2] \leftarrow$ BranchL_KS$(p_\gamma, c, n, W, S)$;
  **14**     $RL \leftarrow$ Map(ReLearn_KS, $PL$);
  **15**     $R[I] \leftarrow$ Reduce(max, $RL$);
  **16**   **return** $R[I]$;

---

*selected projects so far. Initially,* $S = \emptyset$. Extract_KS *is a no-op.* Branch_KS *returns* $[(p, c, n - 1, W, S), (p, c, n - 1, W - c[n], S \cup \{n\})]$, *the two subproblems for whether the* $n^{th}$ *project is selected or not.* isBaseCase *checks if the total cost exceeds the budget or if no projects are left to consider.*

**Limitations of requirements on the recursive algorithm**  Before Example 2, we specified requirements on recursive algorithms considered in this paper. Even though the framework is general, there are algorithms that the framework excludes. Perhaps the most stringent restrictions are on the arithmetics we allow in ReSolve. Using two examples, we illustrate some limits of our framework.

One simple (but perhaps unnatural) problem that the proposed framework cannot handle directly is the following: given $n$ items with unknown "rewards" $r_i$, find the two items with the maximum product of their rewards ($r_i r_j$). Computing the product of rewards involves multiplying (instead of adding) unknown parameters, which our framework cannot handle directly. On the other hand, the problem is equivalent to maximizing the sum of the logarithm of the two rewards. Thus, in this case, a simple reformulation of the problem makes it amenable to Branch & Learn (though whether the learning would work well is a different question, and depends on the data and the parameters).

However, we can make the problem and objective slightly more complicated (and contrived): choose 3 items, and maximize the sum of pairwise products of rewards, i.e. find items $i, j, k$ to maximize $r_i r_j + r_j r_k + r_i r_k$. It is much less obvious how we can find a natural reformulation of the problem that makes it still amenable to Branch & Learn, illustrating some of the limits of the approach.

## 4   The Branch & Learn Framework

The proposed *Branch & Learn (B&L)* framework methodically transforms a recursive algorithm (ReSolve in Algorithm 2) as described in the last section into a Predict+Optimize learning algorithm. In particular, we adapt the recursive algorithm into the ReLearn template of Algorithm 2, for use as the Convert function (line 7)—the intellectual core of the approach—in Algorithm 1.

Recall from Section 2 that, in the context of the coordinate descent algorithm (Algorithm 1), Convert (ReLearn here) takes as input i) the problem $P_\gamma$—constructed from $P$ as well as the current training example $(A, \theta)$—the problem $P$ expressed as a function of the free coefficient coordinate $\gamma$, and ii) the domain $I_0$ of $\gamma$, also constructed from $(A, \theta)$ (for example, to ensure basic properties of parameters such as non-negativity). From these inputs, ReLearn computes a piecewise linear function, mapping intervals (that partition $I_0$ overall) to objectives (computed under the estimated parameters) of estimated optimal solutions using $\gamma$ taking values within that interval. To construct such a ReLearn procedure, we can simply adapt from the corresponding ReSolve algorithm, and we explain each component of ReLearn here.

ReSolve (Algorithm 2) comprises isBaseCase, BaseCaseL, ExtractL, BranchL, Map and Reduce. Since isBaseCase, Map and Reduce do not directly involve any parameters, they are unchanged from ReSolve. As for BaseCaseL, ExtractL and BranchL, they are obtained from ReSolve by replacing all arithmetic operations with their piecewise linear generalizations.

With these component functions, the structure of ReLearn can then be based on that of ReSolve. ReLearn first checks whether the current subproblem is a base case, and if so, returns the corre-

sponding objective under the estimated parameters. Otherwise, it performs the recursion as follows: i) `ExtractL` computes information $T_\gamma$ for deciding which subproblems to branch into, where crucially this information will depend on the parameters and hence also on the free coefficient $\gamma$. Thus the result of `ExtractL` is a data structure mapping each disjoint subinterval $I$ of $I_0$ to some information $T_\gamma[I]$. In `ReLearn`, we denote the set of subintervals as $\mathbb{I}(T_\gamma)$. ii) With the result of `ExtractL`, we iterate over each $I \in \mathbb{I}(T_\gamma)$, and perform the corresponding `BranchL`, `Map` and `Reduce` to get the estimated objective values for each $\gamma \in I$. Note that, since we map `ReLearn` instead of `ReSolve` to $PL$, even within the current interval $I$, the final estimated objective will generally be further subdivided into smaller pieces. Thus, within $I$, it will be piecewise linear function instead of just a linear function. `Reduce`, using the canonical piecewise linear extension $\boxplus$ of $\oplus$, handles the further subdivisions as desired. iii) Finally, the resulting piecewise linear function $R[I]$ is returned.

**Example 3.** *The adapted `ReLearn_KS` in Algorithm 3 solves the parameterized 0-1 knapsack problem with the inputs of the parameterized problem $P_\gamma^i = (p_\gamma, c, n, W, S)$ and an initial domain $I_0$. The estimated profits $p_\gamma$ is an array of linear functions of the free coefficient $\gamma$, and $I_0$ is the initial domain depending on $A^i$ such that $p_\gamma \geq 0$ for all $\gamma \in I_0$. The components of `ReLearn_KS` are essentially the same as those of `ReSolve_KS` (Algorithm 3), except that the addition in line 11 and the max operation in the `Reduce` function are replaced by their canonical extensions.*

We also note that, while it is possible to completely formalize the framework into a syntactic transformation, the correctness (or the required restrictions on `ReSolve`) of the formalization will depend highly on the precise programming language used (e.g. issues of side effect), and require significant low-level work on the level of the grammar of the language, detracting from the key intuitions and principles behind our framework. For this reason, we do not embark on such formalization in this paper. Nonetheless, in the next section, we give several case studies to showcase problems that are amenable to our framework and show how to adapt those recursive/iterative algorithms to `ReLearn` functions, demonstrating the applicability of the proposed framework.

## 5 Case Studies

This section gives several case studies for our framework. We first demonstrate, using the example of the shortest path problem (SPP), how our framework can recover the DP based method of Demirović *et al.* [5]. We then showcase our framework on a more complicated iterative algorithm for solving the capacitated minimum cost flow problem (MCFP). To demonstrate the full recursion generality that our framework can handle, we use it on the tree-search algorithm for the NP-hard problem of minimum cost vertex cover (MCVC). The last example is the multi-stage scheduling (MSS) problem, which can be handled by our framework straightforwardly but not by many other prior Predict+Optimize methods. We describe the problems in the main paper while details are given in Appendix A.

**Shortest Path** Tabular DP can be viewed as an iterative algorithm, computing the subproblem table row by row—the `Branch` operation in `ReSolve` generates a single subproblem $P'$, which is independent of the unknown parameters (only the parameters to $P'$ depend on the unknown parameters). With this perspective, the work of Demirović *et al.* [5] is a special case of our method.

A concrete example is the Bellman-Ford algorithm [2] for SPP in a weighted directed graph with potentially negative weights. We instantiate `ReSolve` for solving SPP to obtain `ReSolve_SPP` as follows. Suppose $V$ is the set of vertices of the graph. The inputs of `ReSolve_SPP` are $(G^c, D, s, t, N)$, where $G^c \in \mathbb{R}^{|V| \times |V|}$ is a matrix with entry $(G^c)_{uv}$ equal to the cost along the directed edge $uv$, $N$ is a counter, $D$ is an array of shortest $(|V| - N - 1)$-hop distances from the source to each vertex, and $s$ and $t$ are the source and terminal. Initially, $N = |V| - 1$ and all elements in $D$ are initialized to a sufficiently large number (signifying infinity), except that $D[s]$ is set to 0. `isBaseCase_SPP` tests whether $N = 0$, and `BaseCase_SPP` returns $D[t]$. `Extract_SPP` computes an array $D'$ where each element $D'[v]$ is computed by iterating over every vertex $u \neq v$ and computing $D'[v] = \min(D[v], D[u] + (G^c)_{uv})$. `Branch_SPP` returns a single subproblem $[P']$ where $P' = (G^c, D', s, t, N - 1)$, on which `ReSolve_SPP` is called recursively.

We consider the Predict+Optimize setting where the edge costs are unknown. In the parameterized problem $P_\gamma$, parameterized by the free coefficient $\gamma$ in the coordinate descent, the edge costs are non-negative parameters represented by linear functions of $\gamma$. Correspondingly, all operations in `ReLearn_SPP` involving edge costs are replaced by their piecewise linear counterparts (Section 2).

`ReLearn_SPP` outputs a piecewise linear function that is the cost of the estimated shortest path under the estimated parameters, which as explained in Section 2 can be used to compute the regret. See Appendix A.1 for the pseudocode of `ReSolve_SPP` and `ReLearn_SPP`.

**Capacitated Minimum Cost Flow**    We now apply the B&L framework to another iteratively solvable problem: MCFP in a directed graph, where at most one edge exists between any two vertices. Each graph edge has a non-negative capacity and a non-negative cost per unit of flow. Given input $K$, we want to find the least cost to route $K$ units of flow from the source $s$ to the terminal $t$. We consider the parameterized problem where only the edge flow costs are unknown.

We instantiate `ReSolve` for solving MCFP using the successive shortest path algorithm [16], to obtain `ReSolve_MCFP`. Suppose the set of vertices is $V$. The inputs of `ReSolve_MCFP` are $(G^p, G^c, F, s, t)$. The two matrices $G^p \in \mathbb{R}_{\geq 0}^{|V| \times |V|}$ and $G^c \in \mathbb{R}_{\geq 0}^{|V| \times |V|}$ represent the edge capacity and the edge (unit) flow cost of the graph respectively. For the successive shortest path algorithm, we preprocess the graph by adding, for every edge $uv$ in $G$, a reverse edge $vu$ with $0$ capacity and the negated cost $-(G^c)_{uv}$. The variable $F \in \mathbb{R}_{\geq 0}^{|V| \times |V|}$ is a matrix representing the flow sent along each edge so far. `isBaseCase_MCFP` tests whether there is no longer a path from $s$ to $t$ with non-zero capacity in $G^p$ or whether the sum of all flows sent to $t$ is no less than $K$, and `BaseCase_MCFP` adds the costs of all flows in $F$, i.e. $R = \sum_{(u,v)} F_{uv} \cdot (G^c)_{uv}$. `Extract_MCFP` computes a path $T$ from $s$ to $t$ with lowest cost (per unit flow) in $G^c$. `Branch_MCFP` updates the capacity graph $G^p$ and the flow graph $F$ given a path $T$. The capacity of each edge in the path $T$ is decreased and the capacities of the reverse edges in $G^p$ are increased by the value of the sending flow. The value of the sending flow is the minimum value between the largest flow value allowed on $T$ and the remaining flow need to be sent to $t$. The flow is also added to $F$. `ReSolve_MCFP` is called recursively on these new $G^p$ and $F$ as well as the original $G^c$, $s$ and $t$.

Correspondingly in `ReLearn_MCFP`, the input is a problem $P_\gamma$ parameterized by the free coefficient $\gamma$, and all edge costs (the unknown parameters) are expressed as linear functions of $\gamma$. The initial domain $I_0$ for $\gamma$ is restricted so that the edge cost estimates are non-negative for all $\gamma \in I_0$. `ExtractL_MCFP` adapts from the Bellman-Ford `ReLearn_SPP` in the previous case study, which computes a piecewise data structure $T_\gamma$ mapping intervals (for $\gamma$) to different shortest paths (in addition to just the path lengths). For each interval $I$ of $T_\gamma$, `Branch_MCFP` constructs a subproblem $P_\gamma'$ by updating $G^p$ using $T_\gamma[I]$. `ReLearn_MCFP` is recursively called on $P_\gamma'$, until the base case is reached. See Appendix A.2 for the pseudocode of `ReSolve_MCFP` and `ReLearn_MCFP`.

**Minimum Cost Vertex Cover**    Our third example is the MCVC problem, where we show how to apply our framework to a (non-degenerate) recursive algorithm (with multiple branches). Given a graph $G = (V, E)$, there is an associated *cost* $c \in \mathbb{R}^{|V|}$ denoting the cost of picking each vertex. The costs are unknown parameters. The goal is to pick a subset of vertices, minimizing the total cost, subject to the constraint that all edges need to be covered, namely at least one of the two vertices on an edge needs to be picked. This problem is relevant in applications such as building public facilities. Consider, for example, the graph being a road network with edge values being traffic flow, and we wish to build speed cameras at intersections with minimum cost, while covering all the roads.

The recursive algorithm `ReSolve_MCVC` takes input $(G, c, \ell, n, chosen)$, where $G, c, \ell$ are as before, $n$ is the number of levels of (binary) recursion remaining and $chosen$ is the current list of chosen vertices. `isBaseCase_MCVC` checks if $n = 0$, and `BaseCase_MCVC` returns the total cost of vertices in $chosen$ if all edges in $G$ are covered, and returns infinity otherwise. `Extract_MCVC` is a no-op. `Branch_MCVC` creates two subproblems $(G, c, \ell, n-1, chosen)$ and $(G, c, \ell, n-1, chosen \cup \{n\})$, i.e. choosing vertex $n$ or not, and `Reduce` takes the min of the two options. Correspondingly, `ReLearn_MCVC` replaces all arithmetic and $\min$ operations by piecewise linear counterparts. See Appendix A.3 for the pseudocode of `ReSolve_MCVC` and `ReLearn_MCVC`.

**Multi-stage Scheduling**    Our last example is the MSS problem, where a set of items needs to be processed by two machines sequentially. Each machine can process only one item at a time, and each item must be processed first by machine 1 and then machine 2. Furthermore, the items must be processed by both machines in the same order. The goal is to find an ordering for processing items that minimizes the total elapsed time. For this problem, the unknown parameters are the individual item processing times on the machines. We note that a host of prior Predict+Optimize methods

cannot be applied to this problem, including the tabular DP method of Demirović *et al.* [5], as well as the recent work of Guler *et al.* [9] and any linear programming based methods which all require a problem formulation with an objective that is *bilinear* in the unknown parameters and the decision variables (that is, a dot product between these two vectors). Our method on the other hand handles this problem relatively straightforwardly by directly adapting Johnson's rule. See Appendix A.4 for details of `ReSolve_MSS` and `ReLearn_MSS` as well as their pseudocode.

## 6 Comparing Branch & Learn with Latest Work

Both Branch & Learn and the work of Guler *et al.* [9] are exact methods for Predict+Optimize, since they both directly optimize the regret, instead of any surrogates. Both also use coordinate-descent as the high-level optimization routine. The main differences lie in the kind of problems that the methods can apply to, as well as the very different techniques used to achieve each coordinate-descent step.

In terms of applicable problems, Branch & Learn applies to all optimization problems solvable by an algorithm fitting the `ReSolve` template in Section 3. On the other hand, [9] applies to problems with an objective that is bilinear in both the decision variables and the unknown parameters—the objective is the inner product between the two vectors. There are problems amenable to Branch & Learn but not [9], and vice versa. Here we can give some intuition about when each method is applicable.

Suppose we have an optimization problem with a finite feasible solution set. For most practical purposes, such a solution set is enumerable by a recursive algorithm (but not always, since finite sets need not be enumerable for computability reasons). If, additionally, the objective function is bilinear as in the assumption of [9], then Branch & Learn would be able to handle such a problem. In this sense, an optimization problem with a finite solution set that is solvable by [9] would also be solvable by Branch & Learn. The multi-stage scheduling problem with unknown per-item/machine processing times is an example problem with no such bilinear-objective representation (unless we use an exponentially large integer linear program to represent the problem, with exponentially many unknown parameters), yet is easy to fit into our framework.

On the other hand, if the solution space is continuous, Branch & Learn might not be able to handle such an optimization problem. The reason is that Branch & Learn (as it is currently formulated) requires explicitly computing the entire empirical regret function, which is possible only if the regret has a finite representation (say, as a piecewise linear function). When the solution space is continuous, the regret might be a general smooth function even if the objective is bilinear. By contrast, [9] never computes the entire regret function explicitly. Instead, it just computes the regret function at query points, and performs numerical optimization on the function (which is convex when the objective is bilinear). In this sense, [9] is more capable than Branch & Learn in these continuous settings.

In terms of techniques, Branch & Learn automatically generates the "transition points" in the piecewise linear functions by essentially simulating the recursive algorithm. This contrasts the approach of [9], where they leverage the convexity structure in the problem arising from their bilinearity assumption on the objective, and they use a numerical optimization approach (essentially a variant of binary/ternary search) to perform a coordinate-descent step.

## 7 Experimental Evaluation

In this section, we compare our proposed B&L method with other methods for Predict+Optimize. We include 4 classical regression methods: linear regression (LR), $k$-nearest neighbors ($k$-NN), classification and regression tree (CART) and random forest (RF) [8]; 3 approximation methods: smart predict then optimize (SPO) [6], quadratic programming task loss (QPTL) [17], and interior point based approach (IntOpt) [12]; and 2 exact methods: SPO tree and SPO forest [7]. We experiment on three optimization problems: MCFP, MCVC and MSS—problems that the previous DP-based method cannot handle—and use both artificial and real-life data on real-life graphs. For MCFP, we use USANet [11] (24 vertices and 43 edges) and GÉANT [10] (40 vertices and 61 edges). For the NP-hard problem of MCVC, we use two smaller graphs from the Survivable Network Design Library [13]: POLSKA (12 vertices and 18 edges) and PDH (11 vertices and 34 edges). In MCFP, the edge costs are unknown, and the capacities are sampled from $[10, 50]$. The flow value is set to 20, and we select random sources and sinks. In MCVC, the edge costs are unknown. In MSS, the processing times for each item and machine are unknown. We vary the number of jobs from 10 to 40.

Table 1: Mean regrets and standard deviations for MCFP with unknown parameters.

| | Artificial Dataset | | | | Real-life Dataset | | | |
|---|---|---|---|---|---|---|---|---|
| | USANet | | GÉANT | | USANet | | GÉANT | |
| Size | 100 | 300 | 100 | 300 | 100 | 300 | 100 | 300 |
| B&L | **1687.10±729.36** | **1699.24±688.96** | **733.78±294.23** | **732.99±270.88** | **141.37±128.52** | **122.66±85.52** | **72.43±63.58** | **68.44±50.76** |
| LR | 1795.02±792.63 | 1749.24±691.16 | 765.13±345.07 | 743.20±289.98 | 177.55±154.53 | 141.00±92.60 | 84.41±68.03 | 88.66±63.21 |
| k-NN | 1783.50±783.84 | 1791.61±715.92 | 801.65±350.87 | 759.87±278.67 | 285.30±206.60 | 223.95±132.56 | 127.73±79.25 | 121.75±48.45 |
| CART | 2529.74±891.06 | 2436.37±922.07 | 1310.53±422.75 | 1260.65±347.26 | 339.41±219.52 | 313.36±175.66 | 157.82±62.13 | 194.62±88.12 |
| RF | 1783.84±750.26 | 1707.96±670.23 | 766.73±322.94 | 771.19±314.24 | 197.25±162.16 | 138.02±87.96 | 88.25±58.49 | 103.92±56.31 |
| SPO | 2193.87±880.07 | 2071.87±739.14 | 1011.65±322.98 | 937.57±252.87 | 204.89±185.62 | 139.55±93.63 | 82.48±71.62 | 84.18±57.95 |
| QPTL | 2220.47±850.36 | 2244.40±854.32 | 1063.96±355.06 | 1093.78±282.53 | 259.94±236.54 | 212.83±172.12 | 84.60±71.89 | 102.30±61.28 |
| IntOpt | 1796.33±756.94 | 1754.77±701.96 | 778.18±303.49 | 762.99±293.06 | 200.69±174.94 | 140.98±88.58 | 82.42±68.32 | 87.61±58.05 |
| SPO Tree | 1743.68±754.79 | 1723.44±695.73 | 1487.71±879.45 | 1499.39±845.48 | 185.68±156.35 | 153.47±106.54 | 143.57±107.88 | 136.16±85.02 |
| SPO Forest | 1777.86±689.84 | 1736.28±691.08 | 745.43±344.07 | 747.60±292.79 | 178.35±144.90 | 145.52±102.28 | 135.98±101.46 | 132.07±77.23 |
| Average TOV | 10825.18±921.41 | 10835.36±1038.05 | 9831.18±3318.39 | 9784.31±3391.45 | 6831.07±1044.27 | 6660.78±872.37 | 6459.94±2049.60 | 6026.91±2049.46 |

Table 2: Mean regrets and standard deviations for MCVC with unknown parameters.

| | Artificial Dataset | | | | Real-life Dataset | | | |
|---|---|---|---|---|---|---|---|---|
| | POLSKA | | PDH | | POLSKA | | PDH | |
| Size | 100 | 300 | 100 | 300 | 100 | 300 | 100 | 300 |
| B&L | **109.45±15.42** | **110.04±6.55** | **50.40±8.60** | **53.71±5.62** | **2.22±1.36** | **2.18±0.49** | **7.57±6.04** | **5.51±2.62** |
| LR | 115.69±15.59 | 116.36±8.60 | 55.15±9.98 | 56.59±6.70 | 3.56±2.07 | 3.09±0.91 | 8.32±5.86 | 6.10±2.67 |
| k-NN | 123.06±17.26 | 116.97±11.26 | 56.58±8.77 | 58.36±6.16 | 5.16±2.13 | 5.02±1.00 | 11.77±7.23 | 9.78±2.91 |
| CART | 117.23±14.88 | 123.61±13.20 | 89.68±15.95 | 89.40±8.91 | 5.47±2.29 | 5.38±1.21 | 16.92±7.64 | 11.70±3.51 |
| RF | 116.38±14.75 | 117.13±11.75 | 58.75±9.88 | 56.84±6.62 | 4.17±1.83 | 3.91±0.92 | 13.45±7.70 | 7.76±3.36 |
| SPO | 117.50±16.94 | 116.72±9.14 | 78.74±14.98 | 69.46±6.63 | 2.94±1.54 | 2.78±0.82 | 10.00±7.97 | 6.58±3.07 |
| QPTL | 118.14±18.07 | 117.52±9.14 | 81.70±14.83 | 79.31±5.72 | 2.58±1.37 | 2.57±0.62 | 10.89±6.49 | 9.28±3.99 |
| IntOpt | 119.52±16.04 | 118.44±9.70 | 69.66±14.43 | 66.12±8.44 | 2.55±1.32 | 2.30±0.51 | 10.93±6.56 | 8.79±3.93 |
| SPO Tree | 118.90±17.62 | 117.55±9.49 | 56.51±8.95 | 57.70±6.77 | 2.68±1.31 | 2.57±0.65 | 14.12±6.73 | 11.19±4.13 |
| SPO Forest | 119.00±18.72 | 117.25±10.95 | 55.07±8.98 | 56.85±6.00 | 2.86±1.42 | 2.66±0.64 | 13.11±7.40 | 11.45±3.62 |
| Average TOV | 649.00±21.02 | 654.82±11.83 | 817.76±26.29 | 822.72±16.22 | 321.14±16.11 | 317.96±6.96 | 502.17±30.42 | 503.57±12.39 |

We run 30 simulations for each problem configuration. In each simulation, we build datasets consisting of $n \in \{100, 300\}$ pairs of (feature matrix, parameters). In the artificial and real-life datasets, each parameter has 4 and 8 features respectively. Given that we are unable to find datasets specifically for the MCFP, MCVC and MSS problems, we follow the experimental approach of Demirović *et al.* [3, 4, 5] and use real data from a different problem (the ICON scheduling competition) as numerical values required for our experiment instances. We use a 70%/30% training/testing data split. Details of the data generation method are in Appendix B. We use the *scikit-learn* library [1] to implement LR, k-NN, CART and RF, and *OR-Tools* [14] as the problem solver in SPO Tree and SPO Forest. All models are trained with Intel(R) Xeon(R) CPU E5-2630 v2 @ 2.60GHz processors.[1]

**Solution Quality** Table 1, 2, and 3 report the mean regrets and their standard deviations for each method on MCFP, MCVC and MSS respectively. *Mean regret ± std* is the metric used to demonstrate the performance. At the bottom of the table we also report the *average true optimal values* (TOV) to compare the relative error on the artificial and real-life datasets.

As shown in Table 1, B&L achieves the best performance in all cases. On the artificial dataset, B&L, LR, k-NN, RF, IntOpt, and SPO Forest achieve similar performance on both of the two graphs, while SPO Tree performs well on USANet but has a poor performance on GÉANT. With the real-life dataset, B&L shows the most significant advantages. Compared with other methods, B&L obtains 20.38%-58.35% ($n = 100$) and 11.13%-60.86% ($n = 300$) smaller regret on USANet, and 12.13%-54.11% ($n = 100$) and 18.70%-64.83% ($n = 300$) smaller regret on GÉANT. We observe that all methods achieve smaller relative error with real-life data than with artificial data. B&L, for example, achieves 7.46%-15.68% and 1.12%-2.07% relative error with artificial and real-life data respectively. This is consistent with how the artificial dataset is purposefully designed to be highly non-linear (see Appendix B), and thus harder to learn. Nevertheless, B&L still achieves the smallest regret.

Table 2 shows the results for MCVC. We can see that B&L has the smallest mean regrets in all cases. On the artificial dataset, all methods achieve similar good performance on POLSKA, while CART and all approximation methods perform poorly on PDH. The performance differences among different methods are larger in the real-life dataset, and the advantages of B&L are more evident. B&L obtains 12.95%-59.41% ($n = 100$) and 5.22%-59.53% ($n = 300$) smaller regret in POLSKA, and 9.05%-55.26% ($n = 100$) and 9.71%-52.89% ($n = 300$) in PDH. Similar to the results of MCFP, all algorithms achieve better performance in the real-life dataset. B&L achieves 6.16%-16.86% relative error in the artificial dataset, and 0.68%-1.51% relative error in the real-life dataset.

---

[1]Our implementation is available at https://github.com/dadahxy/NeurIPS_BranchAndLearn

Table 3: Mean regrets and standard deviations for MSS with unknown parameters.

| | Artificial Dataset | | | | Real-life Dataset | | | |
|---|---|---|---|---|---|---|---|---|
| | 10 Jobs | | 40 Jobs | | 10 Jobs | | 40 Jobs | |
| Size | 100 | 300 | 100 | 300 | 100 | 300 | 100 | 300 |
| B&L | **112.23±10.74** | **118.03±6.40** | **216.23±18.10** | **220.88±12.25** | **2.63±1.25** | **2.41±0.57** | **18.52±2.98** | **21.17±1.55** |
| LR | 121.91±12.12 | 121.53±6.66 | 226.74±21.34 | 225.80±16.28 | 3.08±1.37 | 2.82±0.58 | 20.78±3.65 | 23.48±2.04 |
| $k$-NN | 122.67±14.31 | 123.89±6.98 | 227.95±28.40 | 221.45±12.62 | 7.88±2.14 | 7.46±0.93 | 54.24±7.81 | 48.23±4.17 |
| CART | 117.79±14.46 | 123.67±7.41 | 225.29±24.91 | 227.99±12.45 | 8.53±2.23 | 6.43±0.98 | 52.45±8.86 | 35.19±5.50 |
| RF | 120.72±12.38 | 123.90±6.18 | 223.29±22.52 | 228.13±16.86 | 6.92±2.02 | 5.71±1.00 | 44.52±6.61 | 31.01±4.24 |
| SPO Tree | 122.21±16.07 | 122.23±7.81 | 231.14±30.03 | 229.43±17.39 | 4.53±1.53 | 4.19±0.83 | 49.21±9.67 | 42.98±5.59 |
| SPO Forest | 125.37±12.52 | 121.64±6.76 | 223.20±22.00 | 230.55±12.96 | 3.38±1.40 | 3.31±0.69 | 46.18±10.06 | 35.59±3.25 |
| Average TOV | 1215.19±25.89 | 1223.62±16.46 | 4606.86±57.62 | 4599.35±38.33 | 548.99±19.79 | 550.05±13.59 | 2644.11±52.04 | 2671.86±34.35 |

Table 4: Average runtime (in seconds) for MCFP, MCVC and MSS on real-life data.

| | Minimum cost flow problem | | | | Minimum cost vertex covering problem | | | | Multi-stage scheduling problem | | | |
|---|---|---|---|---|---|---|---|---|---|---|---|---|
| | USANet | | GÉANT | | POLSKA | | PDH | | 10 Jobs | | 40 Jobs | |
| Size | 100 | 300 | 100 | 300 | 100 | 300 | 100 | 300 | 100 | 300 | 100 | 300 |
| B&L | 22.47 | 70.37 | 20.63 | 59.17 | 651.76 | 1965.00 | 298.26 | 896.00 | 8.02 | 27.74 | 1427.34 | 4302.60 |
| SPO Tree | 20.94 | 61.93 | 35.94 | 439.19 | 484.77 | 6281.35 | 223.53 | 2798.22 | 7.88 | 41.80 | 156.56 | 4683.33 |
| SPO Forest | 17.26 | 73.05 | 22.07 | 369.21 | 980.32 | 4277.86 | 488.36 | 2125.27 | 11.03 | 49.17 | 66.05 | 2424.03 |

We show the mean regrets and standard deviations in the MSS experiment in Table 3. Note that linear programming based methods including SPO, QPTL and IntOpt are not applicable to this problem as mentioned in Section 5. Due to the space limitation, we only show the results when the number of jobs is 10 and 40—see Appendix C for more results. B&L has the best performance in all cases with the real-life dataset, achieving 0.44%-0.79% relative error compared to the TOV. Contrasting other methods, B&L obtains 14.58%-69.20% ($n = 100$) and 14.47%-67.73% ($n = 300$) smaller regret for 10 jobs, and 10.85%-65.85% ($n = 100$) and 9.84%-56.10% ($n = 300$) for 40 jobs. On the artificial dataset, all algorithms perform essentially the same, achieving 4.69%-9.65% relative error.

**Scalability/Runtime**   Learning using regression or approximate methods is fast, but these methods sacrifice the accuracy of the learned model, which is the motivation for the line of work on Predict+Optimize. On the other hand, many optimization problems are expensive to solve. Although exact methods achieve lower regret, their runtime scalability can be an issue since their learning process requires solving optimization problems multiple times. For a fair comparison, therefore, we compare B&L only with other exact methods: SPO Tree and SPO Forest. Table 4 shows the average runtime across 30 simulations for different cases. Overall, we observe that B&L scales at least as well as SPO Tree and SPO Forest in most of the cases. Of note is the (MSS, 40 jobs, dataset size 100) setting where B&L runs slower than SPO Tree and Forest, due to the fact that MSS is a permutation-based scheduling problem, where B&L takes significant training time to explore all the possible predictions. Nonetheless, if we consider the ratio of runtimes between dataset sizes 100 and 300, B&L scales significantly better than SPO Tree and Forest with respect to the dataset size.

# 8   Summary

Given a `ReSolve` function for a recursively or iteratively solvable problem, we propose a systematic approach to synthesize a `ReLearn` function for learning with the regret loss, by replacing operators in `ReSolve` with their piecewise linear counterparts. Our proposal is methodical and straightforward to implement. Furthermore, our framework encompasses a wide class of recursive algorithms (`ReSolve` functions), as demonstrated by our case studies. Most importantly, B&L empirically achieves the lowest regrets against classical machine learning and contemporary Predict+Optimize algorithms with runtime comparable with the latter.

# Acknowledgments

We thank the anonymous referees for their constructive comments. In addition, we acknowledge the financial support of a General Research Fund (RGC Ref. No. CUHK 14206321) by the University Grants Committee, Hong Kong. Jasper C.H. Lee was supported in part by the generous funding of a Croucher Fellowship for Postdoctoral Research, NSF award DMS-2023239, NSF Medium Award CCF-2107079 and NSF AiTF Award CCF-2006206.

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
