# A    Pseudocode for Case Studies

This section gives pseudocode implementations for the recursive and iterative algorithmic case studies in the main paper, which includes **i)** the Bellman-Ford algorithm for the shortest path problem (Algorithms 4 and 5), **ii)** the successive shortest path algorithm for the minimum cost flow problem (Algorithm 6 and 7), **iii)** the branching algorithm for the minimum cost vertex covering problem (Algorithm 8 and 9), and **iv)** the Johnson's rule for the multi-stage scheduling problem (Algorithm 10 and 11).

Note that for the Bellman-Ford and successive shortest path algorithms, `Branch` only generates one subproblem. Thus it is equivalent to invoking the recursive call on the subproblem directly without calling the `Map` function.

## A.1    Pseudocode for the Bellman-Ford Algorithm for Shortest Path

---

**Algorithm 4:** Bellman-Ford Algorithm for SPP

---

1 **Function** $ReSolve\_SPP(G^c, D, s, t, N)$**:**
2     **if** $N = 0$ **then**
3         $R \leftarrow D[t]$;
4     **else**
5         $D' \leftarrow \texttt{Extract\_SPP}(G^c, D)$;
6         $[P'] \leftarrow \texttt{Branch\_SPP}(G^c, D', s, t, N-1)$;
7         $R \leftarrow \texttt{ReSolve\_SPP}(P')$;
8     **return** $R$;
9 **Function** $Extract\_SPP(G^c, D)$**:**
10     **for** *every vertex $u \neq v$ in $V$* **do**
11         $D'[v] = \min(D[v], D[u] + (G^c)_{uv})$;
12     **return** $D'$

---

---

**Algorithm 5:** Bellman-Ford Algorithm for Para-SPP

---

1 **Function** $ReLearn\_SPP(G^c_\gamma, D_\gamma, s, t, N, I_0)$**:**
2     **if** $N = 0$ **then**
3         $R[I_0] \leftarrow D_\gamma[t]$;
4     **else**
5         $D'_\gamma \leftarrow \texttt{ExtractL\_SPP}(G^c_\gamma, D_\gamma, I_0)$;
6         $[P'] \leftarrow \texttt{BranchL\_SPP}(G^c_\gamma, D'_\gamma, s, t, N-1)$;
7         $R[I_0] \leftarrow \texttt{ReLearn\_SPP}(P', I_0)$;
8     **return** $R$;
9 **Function** $ExtractL\_SPP(G^c_\gamma, D_\gamma, I_0)$**:**
10     **for** *every vertex $u \neq v$ in $V$* **do**
11         $D'_\gamma[v] = \min(D_\gamma[v], D_\gamma[u] + (G^c_\gamma)_{uv})$;
12     **return** $D'_\gamma$

---

## A.2 Pseudocode for the Successive Shortest Path Algorithm for Capacitated Minimum Cost Flow

---

**Algorithm 6:** Successive Shortest Path Algorithm for MCFP

---

**1 Function** $ReSolve\_MCFP(G^p, G^c, F, s, t)$**:**

**2**    **if** *no path from s to t in $G^p$ or* $\sum_{(u,t)} F_{ut} \geq K$ **then**

**3**      $R \leftarrow \sum_{(u,v)} F_{uv} \cdot (G^c)_{uv}$;

**4**    **else**

**5**      $T \leftarrow \texttt{Extract\_MCFP}(G^p, G^c, s, t)$;

**6**      $[P'] \leftarrow \texttt{Branch\_MCFP}(G^p, G^c, F, T)$;

**7**      $R \leftarrow \texttt{ReSolve\_MCFP}(P')$;

**8**    **return** $R$;

**9 Function** $Extract\_MCFP(G^p, G^c, s, t)$**:**

**10**    **for** *every vertex $v$ in $V$* **do**

**11**      **for** *every vertex $u \neq v$ in $V$* **do**

**12**        **if** $(G^p)_{uv} > 0$ **then**

**13**          $D[v] \leftarrow \min(D[v], D[u] + (G^c)_{uv})$;

**14**          **if** $D[v] = D[u] + (G^c)_{uv}$ **then**

**15**            $T[v] \leftarrow u$;

**16**    **return** $T$;

**17 Function** $Branch\_MCFP(G^p, G^c, F, T)$**:**

**18**    $block\_flow = \infty$;

**19**    **for** *every vertex $v$ on $T[\,]$* **do**

**20**      $u \leftarrow T[v]$;

**21**      $block\_flow \leftarrow \min(block\_flow, F_{uv})$;

**22**    $block\_flow \leftarrow \min(block\_flow, K - \sum_{(u,t)} F_{ut})$;

**23**    **for** *every vertex $v$ on $T[\,]$* **do**

**24**      $u \leftarrow T[v]$;

**25**      $(G^{p'})_{uv} \leftarrow (G^p)_{uv} - block\_flow$;

**26**      $(G^{p'})_{vu} \leftarrow (G^p)_{vu} + block\_flow$;

**27**      $F'_{uv} \leftarrow F_{uv} + block\_flow$;

**28**    **return** $(G^{p'}, G^c, F', s, t)$;

---

**Algorithm 7:** Successive Shortest Path Algorithm for Para-MCFP

**1 Function** $ReLearn\_MCFP(G^p, G^c_\gamma, F_\gamma, s, t, I_0)$:

**2**     **if** *no path from s to t in $G^p$ or* $\sum_I \sum_{(u,t)} (F_\gamma[I])_{ut} \geq K$ **then**

**3**         $R[I_0] \leftarrow \sum_I \sum_{(u,v)} (F_\gamma[I])_{uv} \cdot (G^c_\gamma)_{uv}$;

**4**     **else**

**5**         $T_\gamma \leftarrow \texttt{ExtractL\_MCFP}(G^p, G^c_\gamma, s, t, I_0)$;

**6**         **for** *each interval $I \in \mathbb{I}(T_\gamma)$* **do**

**7**             $[P'] \leftarrow \texttt{BranchL\_MCFP}(G^p, G^c_\gamma, F_\gamma[I], T_\gamma[I])$;

**8**             $R[I] \leftarrow \texttt{ReLearn\_MCFP}(P', I)$;

**9**     return $R[I_0]$;

**10 Function** $ExtractL\_MCFP(G^p, G^c_\gamma, s, t, I_0)$:

**11**     **for** *every vertex $v$ in $V$* **do**

**12**         **for** *every vertex $u \neq v$ in $V$* **do**

**13**             **if** $(G^p)_{uv} > 0$ **then**

**14**                 $D_\gamma[v] \leftarrow \min(D_\gamma[v], D_\gamma[u] + (G^c_\gamma)_{uv})$;

**15**                 **for** *each interval $I \in \mathbb{I}(D'_\gamma)$* **do**

**16**                     **if** $D_\gamma[v] = D_\gamma[u] + (G^c_\gamma)_{uv}$ **then**

**17**                         $T_\gamma[I][v] \leftarrow u$;

**18**     return $T_\gamma$;

**19 Function** $BranchL\_MCFP(G^p, G^c_\gamma, F_\gamma[I], T_\gamma[I])$:

**20**     $block\_flow \leftarrow \infty$;

**21**     **for** *every vertex $v$ on $T_\gamma[I][]$* **do**

**22**         $u \leftarrow T_\gamma[I][v]$;

**23**         $block\_flow \leftarrow \min(block\_flow, (F_\gamma[I])_{uv})$;

**24**     $block\_flow \leftarrow \min(block\_flow, K - \sum_I \sum_{(u,t)} (F_\gamma[I])_{ut})$;

**25**     **for** *every vertex $v$ on $T_\gamma[I][]$* **do**

**26**         $u \leftarrow T_\gamma[I][v]$;

**27**         $(G^{p\prime})_{uv} \leftarrow (G^p)_{uv} - block\_flow$;

**28**         $(G^{p\prime})_{vu} \leftarrow (G^p)_{vu} + block\_flow$;

**29**         $F'_\gamma[I]_{uv} \leftarrow (F_\gamma[I])_{uv} + block\_flow$;

**30**     return $(G^{p\prime}, G^c_\gamma, F'_\gamma[I], s, t, I)$;

## A.3   Pseudocode for the standard Branching Algorithm for Minimum Cost Vertex Cover

---

**Algorithm 8:** Branching Algorithm for MCVC

---

**1  Function** $ReSolve\_MCVC(G, c, \ell, n, chosen)$**:**
**2**    **if** $n = 0$ **then**
**3**        **if** *the edges in G are all covered* **then**
**4**            $R \leftarrow 0$;
**5**            **for** *every vertex v in chosen[ ]* **do**
**6**                $R \leftarrow R + c[v]$;
**7**            **return** $R$;
**8**        **else**
**9**            **return** $\infty$;
**10**    **else**
**11**        $[P_1, P_2] \leftarrow \texttt{Branch\_MCVC}(G, c, \ell, n, chosen)$;
**12**        $[R_1, R_2] \leftarrow \texttt{Map}(\texttt{ReSolve\_MCVC}, [P_1, P_2])$;
**13**        $R \leftarrow \texttt{Reduce}(\boldsymbol{min}, [R_1, R_2])$;
**14**    **return** $R$;
**15  Function** $Branch\_MCVC(G, c, \ell, n, chosen)$**:**
**16**    $P_1 \leftarrow (G, c, \ell, n - 1, chosen)$;
**17**    $P_2 \leftarrow (G, c, \ell, n - 1, chosen \cup \{n\})$;
**18**    **return** $(P_1, P_2)$;

---

---

**Algorithm 9:** Branching Algorithm for Para-MCVC

---

**1  Function** $ReLearn\_MCVC(G, c_\gamma, \ell_\gamma, n, chosen_\gamma, I_0)$**:**
**2**    **if** $n = 0$ **then**
**3**        **if** *the edges in G are all covered* **then**
**4**            $R[I_0] \leftarrow 0$;
**5**            **for** *every vertex v in* $chosen_\gamma[\,]$ **do**
**6**                $R[I_0] \leftarrow R[I_0] + c_\gamma[v]$;
**7**            **return** $R[I_0]$;
**8**        **else**
**9**            $R[I_0] \leftarrow \infty$;
**10**            **return** $R[I_0]$;
**11**    **else**
**12**        $[P_1, P_2] \leftarrow \texttt{BranchL\_MCVC}(G, c_\gamma, \ell_\gamma, n, chosen_\gamma, I_0)$;
**13**        $[R_1, R_2] \leftarrow \texttt{Map}(\texttt{ReLearn\_MCVC}, [P_1, P_2])$;
**14**        $R \leftarrow \texttt{Reduce}(\boldsymbol{min}, [R_1, R_2])$;
**15**    **return** $R$;
**16  Function** $BranchL\_MCVC(G, c_\gamma, \ell_\gamma, n, chosen_\gamma, I_0)$**:**
**17**    $P_1 \leftarrow (G, c_\gamma, \ell_\gamma, n - 1, chosen_\gamma, I_0)$;
**18**    $P_2 \leftarrow (G, c_\gamma, \ell_\gamma, n - 1, chosen_\gamma \cup \{n\}, I_0)$;
**19**    **return** $(P_1, P_2)$;

---

## A.4   Pseudocode for the Johnson's rule for Multi-stage Scheduling

Suppose $schedule$ is the current array of the scheduled items which is initialized as an array of length equal to the total number of items and all elements equal to -1, $remainingItems$ is the current list of the unscheduled items, and $timeReq$ is the array with two columns, each containing the processing times of each item on a machine. Throughout the execution of the algorithm, the form of $schedule$ always maintains the structure of $[a, b, c]$ where $a$ is a sub-array of non-$(-1)$ entries corresponding to actual items, $b$ is a sub-array of only $-1$ entries, and $c$ is another sub-array

of non-$(-1)$ entries corresponding to actual items. The recursive algorithm `ReSolve_MSS` takes input $(n, schedule, remainingItems, timeReq)$, where $n$ is the remaining number of levels of recursion. `isBaseCase_MSS` checks if $n = 0$, and `BaseCase_MSS` returns the total elapsed time for processing all items on both of the two machines. `Extract_MSS` examines all the entries of $timeReq$ that correspond to items in $remainingItems$, and finds the smallest entry. This entry corresponds to an $(item, machine)$ pair, where machine can be either 1 or 2, and this pair is returned by `Extract_MSS`. Given this as input, `Branch_MSS` first removes the item from $remainingItems$, and needs to now insert this item into $schedule$. The way it does that is simple: if the entry found by `Extract_MSS` corresponds to machine 1, it looks for the first -1 entry in $schedule$ from the beginning of the array and replaces it with $item$. Otherwise, it instead looks from the end of the array backwards for the first -1 entry, and replaces it with $item$. Given this processing, `Branch_MSS` returns the single subproblem $(n-1, schedule, remainingItems, timeReq)$ and `ReSolve_MSS` is called recursively.

Correspondingly, the input of `ReLearn_MSS` is a problem $P_\gamma$ parameterized by the free coefficient $\gamma$, and the processing time of all items on two machines (the unknown parameters) are expressed as linear functions of $\gamma$. `ExtractL_MSS` replaces all arithmetic and $\min$ operations by piecewise linear counterparts, which computes a piecewise data structure $T_\gamma$ mapping intervals (for $\gamma$) to different inserted items that with the shortest processing time in the $remainingItems$ list and different corresponding machine numbers. For each interval $I$ of $T_\gamma$, `Branch_MSS` constructs a subproblem $P'_\gamma$ by updating $schedule$ and $remainingItems$ using $T_\gamma[I]$. `ReLearn_MSS` is recursively called on $P'_\gamma$, until the base case is reached.

---
**Algorithm 10:** Johnson's Rule for MSS
---

**1 Function** $ReSolve\_MSS(n, schedule, remainingItems, timeReq)$**:**

**2**    **if** $n = 0$ **then**

**3**       $M1Time \leftarrow 0$;

**4**       $M2Time \leftarrow 0$;

**5**       **for** *every item i in* $schedule[]$ **do**

**6**          $M1Time \leftarrow M1Time + timeReq[i][1]$;

**7**          **if** $M1Time < M2Time$ **then**

**8**             $M2Time \leftarrow M2Time + timeReq[i][2]$;

**9**          **else**

**10**             $M2Time \leftarrow M1Time + timeReq[i][2]$;

**11**       $R \leftarrow M2Time$;

**12**    **else**

**13**       $T \leftarrow \text{Extract\_MSS}(remainingItems, timeReq)$;

**14**       $[P'] \leftarrow \text{Branch\_MSS}(n, schedule, remainingItems, timeReq, T)$;

**15**       $R \leftarrow \text{ReSolve\_MSS}(P')$;

**16**    return $R$;

**17 Function** $Extract\_MSS(remainingItems, timeReq)$**:**

**18**    $minProTime \leftarrow \infty$;

**19**    $item \leftarrow -1$;

**20**    $machine \leftarrow -1$;

**21**    **for** *every item i in* $remainingItems[]$ **do**

**22**       $minProTime \leftarrow \min(minProTime, timeReq[i][1], timeReq[i][2])$;

**23**       **if** $minProTime = timeReq[i][1]$ **then**

**24**          $item \leftarrow i$;

**25**          $machine \leftarrow 1$;

**26**       **if** $minProTime = timeReq[i][2]$ **then**

**27**          $item \leftarrow i$;

**28**          $machine \leftarrow 2$;

**29**    $T \leftarrow (item, machine)$;

**30**    return $T$;

**31 Function** $Branch\_MSS(n, schedule, remainingItems, timeReq, T)$**:**

**32**    Remove $T.item$ from $remainingItems[]$;

**33**    **if** $T.machine = 1$ **then**

**34**       Look from the beginning of $schedule[]$ forward for the first -1 entry, and replace it with $T.item$;

**35**    **else**

**36**       Look from the end of $schedule[]$ backward for the first -1 entry, and replace it with $T.item$;

**37**    return $(n - 1, schedule, remainingItems, timeReq)$;

---

**Algorithm 11:** Johnson's Rule for Para-MSS

---

**1** **Function** $ReLearn\_MSS(n, schedule_\gamma, remainingItems_\gamma, timeReq_\gamma, I_0)$**:**

**2**     **if** $n = 0$ **then**

**3**         $M1Time \leftarrow 0$;

**4**         $M2Time \leftarrow 0$;

**5**         **for** *every item i in* $schedule_\gamma[]$ **do**

**6**             $M1Time \leftarrow M1Time + timeReq_\gamma[i][1]$;

**7**             **if** $M1Time < M2Time$ **then**

**8**                 $M2Time \leftarrow M2Time + timeReq_\gamma[i][2]$;

**9**             **else**

**10**                 $M2Time \leftarrow M1Time + timeReq_\gamma[i][2]$;

**11**         $R[I_0] \leftarrow M2Time$;

**12**     **else**

**13**         $T_\gamma \leftarrow \texttt{ExtractL\_MSS}(remainingItems_\gamma, timeReq_\gamma, I_0)$;

**14**         **for** *each interval* $I \in \mathbb{I}(T_\gamma)$ **do**

**15**             $[P'] \leftarrow$

                $\texttt{BranchL\_MSS}(n, schedule_\gamma[I], remainingItems_\gamma[I], timeReq_\gamma, T_\gamma[I])$;

**16**             $R[I] \leftarrow \texttt{ReLearn\_MSS}(P', I)$;

**17**     return $R[I_0]$;

**18** **Function** $ExtractL\_MSS(remainingItems_\gamma, timeReq_\gamma, I_0)$**:**

**19**     $minProTime_\gamma \leftarrow \infty$;

**20**     $item \leftarrow -1$;

**21**     $machine \leftarrow -1$;

**22**     **for** *every item i in* $remainingItems_\gamma[]$ **do**

**23**         $minProTime_\gamma \leftarrow \min(minProTime, timeReq_\gamma[i][1], timeReq_\gamma[i][2])$;

**24**         **for** *each interval* $I \in \mathbb{I}(minProTime_\gamma)$ **do**

**25**             **if** $minProTime_\gamma[I] = timeReq_\gamma[i][1]$ **then**

**26**                 $T_\gamma[I].item \leftarrow i$;

**27**                 $T_\gamma[I].machine \leftarrow 1$;

**28**             **if** $minProTime_\gamma[I] = timeReq_\gamma[i][2]$ **then**

**29**                 $T_\gamma[I].item \leftarrow i$;

**30**                 $T_\gamma[I].machine \leftarrow 2$;

**31**     return $T_\gamma$;

**32** **Function** $BranchL\_MSS(n, schedule_\gamma[I], remainingItems_\gamma[I], timeReq_\gamma, T_\gamma[I])$**:**

**33**     Remove $T_\gamma[I].item$ from $remainingItems_\gamma[I]$;

**34**     **if** $T_\gamma[I].machine = 1$ **then**

**35**         Look from the beginning of $schedule_\gamma[I]$ forward for the first -1 entry, and replace it with $T_\gamma[I].item$;

**36**     **else**

**37**         Look from the end of $schedule_\gamma[I]$ backward for the first -1 entry, and replace it with $T_\gamma[I].item$;

**38**     return $(n-1, schedule_\gamma[I], remainingItems_\gamma[I], timeReq_\gamma)$;

---

# B   Experimental Setting

In this section, we give details about the experimental setting, including the random source and sink selection method used in MCFP, the generation of the artificial dataset for each problem we experiment on, and the parameters tuning.

**The Random Source and Sink Selection Method Used in MCFP**   We design graph-specific distributions for taking a random source and a random sink for the minimum cost flow problem (MCFP), with the goal of making sure that the path between the source and the sink is not too short

Table 5: Mean regrets and standard deviations for MSS with unknown parameters in artificial dataset.

| Size | 10 Jobs | | 20 Jobs | | 30 Jobs | | 40 Jobs | |
|---|---|---|---|---|---|---|---|---|
| | 100 | 300 | 100 | 300 | 100 | 300 | 100 | 300 |
| B&L | **112.23±10.74** | **118.03±6.40** | **162.05±11.90** | **164.73±6.74** | **189.70±17.62** | **196.33±8.90** | **216.23±18.10** | **220.88±12.25** |
| LR | 121.91±12.12 | 121.53±6.66 | 168.03±14.47 | 171.71±8.09 | 201.32±21.26 | 204.38±10.31 | 226.74±21.34 | 225.80±16.28 |
| $k$-NN | 122.67±14.31 | 123.89±6.98 | 168.52±18.96 | 170.40±11.08 | 201.88±22.63 | 203.85±11.74 | 227.95±28.40 | 221.45±12.62 |
| CART | 117.79±14.46 | 123.67±7.41 | 167.56±22.89 | 170.08±9.13 | 200.31±23.75 | 202.04±12.57 | 225.29±24.91 | 227.99±12.45 |
| RF | 120.72±12.38 | 123.90±6.18 | 172.18±20.69 | 171.02±11.81 | 205.62±20.37 | 205.04±11.25 | 223.29±22.52 | 228.13±16.86 |
| SPO Tree | 122.21±16.07 | 122.23±7.81 | 175.15±19.87 | 170.29±11.96 | 203.94±21.78 | 204.88±11.09 | 231.14±30.03 | 229.43±17.39 |
| SPO Forest | 125.37±12.52 | 121.64±6.76 | 171.11±18.06 | 171.17±11.91 | 208.32±23.14 | 206.16±16.65 | 223.20±22.00 | 230.55±12.96 |
| Average TOV | 1215.19±25.89 | 1223.62±16.46 | 2351.32±34.52 | 2347.40±23.01 | 3481.60±55.41 | 3473.42±30.45 | 4606.86±57.62 | 4599.35±38.33 |

Table 6: Mean regrets and standard deviations for MSS with unknown parameters in real-life dataset.

| Size | 10 Jobs | | 20 Jobs | | 30 Jobs | | 40 Jobs | |
|---|---|---|---|---|---|---|---|---|
| | 100 | 300 | 100 | 300 | 100 | 300 | 100 | 300 |
| B&L | **2.63±1.25** | **2.41±0.57** | **7.91±2.02** | **7.85±1.34** | **14.55±3.34** | **15.24±2.34** | **18.52±2.98** | **21.17±1.55** |
| LR | 3.08±1.37 | 2.82±0.58 | 9.82±2.06 | 9.35±1.52 | 17.87±3.22 | 17.38±2.33 | 20.78±3.65 | 23.48±2.04 |
| $k$-NN | 7.88±2.14 | 7.46±0.93 | 29.76±6.07 | 25.70±3.38 | 46.56±7.57 | 39.84±3.80 | 54.24±7.81 | 48.23±4.17 |
| CART | 8.53±2.23 | 6.43±0.98 | 24.80±7.15 | 21.86±3.55 | 44.77±8.68 | 33.02±4.51 | 52.45±8.86 | 35.19±5.50 |
| RF | 6.92±2.02 | 5.71±1.00 | 22.18±5.24 | 19.94±3.04 | 36.40±6.25 | 28.37±3.31 | 44.52±6.61 | 31.01±4.24 |
| SPO Tree | 4.53±1.53 | 4.19±0.83 | 13.51±4.84 | 12.12±3.15 | 43.39±8.30 | 36.89±4.24 | 49.21±9.67 | 42.98±5.59 |
| SPO Forest | 3.38±1.40 | 3.31±0.69 | 11.96±4.87 | 10.20±1.90 | 46.02±7.86 | 33.83±3.53 | 46.18±10.06 | 35.59±3.25 |
| Average TOV | 548.99±19.79 | 550.05±13.59 | 1377.53±53.51 | 1394.51±24.25 | 2040.78±55.05 | 2022.41±34.75 | 2644.11±52.04 | 2671.86±34.35 |

(e.g. length 1). In USANet, we randomly choose the source from vertices $\{1, 2, 3, 4, 5\}$ and the sink from vertices $\{20, 21, 22, 23, 24\}$. In GÉANT, the source and sink are randomly selected from all the points with zero in-degree and zero out-degree respectively.

**The Generation of the Artificial Dataset** The artificial datasets for the three problems are generated as follows. Each feature is a 4-tuple $\vec{a}_{uv} = (a_{uv1}, a_{uv2}, a_{uv3}, a_{uv4})$, where $a_{uv1} \in \{1, 2, \ldots, 7\}$ represents the day of the week, $a_{uv2} \in \{1, 2, \ldots, 30\}$ represents the day of the month, and $a_{uv3}, a_{uv4} \in [0, 360]$ represent the meteorology index and road congestion respectively. The true parameters are generated by $10 * sin(a_{uv1}) * sin(a_{uv2}) + 100 * sin(a_{uv3}) * sin(a_{uv4}) + C$, where $C$ is a large positive constant to ensure the value of each learned parameter is positive. We use such nonlinear mapping to compare the performance of our proposed methods and that of other methods.

**Parameters Tuning** As for the parameters tuning, we try different settings for k-nearest neighbors (k-NN), Random forest (RF), SPO tree (SPOT), and SPO Forest. In k-NN, we try the regression model with $k \in \{1, 3, 5\}$. As for RF, we try different numbers of trees in the forest $n\_estimator \in \{10, 50, 100\}$. We tune the maximum depths of the tree $max\_depth \in \{1, 3, 10, 100\}$ and the minimum weights per node $min\_weights \in \{5, 20, 30\}$ for SPOT. For SPO Forest, we tune two parameters: the maximum depth of each tree $max\_depth \in \{1, 3, 10, 100\}$ and the number of trees in the forest $n\_estimator \in \{10, 50, 100\}$.

We end this appendix section with a remark on our experiments using real-life data. Given that we are unable to find datasets specifically for the MCFP, MCVC and MSS problems, we follow the experimental approach of Demirović *et al.* [3, 4, 5] and use real data from a different problem (the ICON scheduling competition) as numerical values required for our experiment instances.

## C   Additional Results: Multi-stage Scheduling

This section shows the experiment results of the job number of 10, 20, 30, and 40 in the MSS experiment with artificial and real-life datasets.

Table 5 reports the mean regrets and their standard deviations for each method on the artificial dataset. Although all algorithms achieve similar performances, B&L performs (slightly) better than all other methods in all cases. The results on the real-life dataset are shown in Table 6. Compared with the artificial dataset, the performance differences among different methods are larger in the real-life dataset, and the advantages of B&L are more evident. Contrasting other methods, B&L obtains 14.58%-69.20% ($n = 100$) and 14.47%-67.73% ($n = 300$) smaller regret for 10 jobs, 19.45%-73.41% ($n = 100$) and 16.09%-69.45% ($n = 300$) smaller regret for 20 jobs, 18.56%-68.74% ($n = 100$) and 12.29%-61.74% ($n = 300$) smaller regret for 30 jobs, and 10.85%-65.85% ($n = 100$) and 9.84%-56.10% ($n = 300$) for 40 jobs. Similar to the results of MCFP and MCVC, all algorithms achieve better performance on the real-life dataset. For example, B&L achieves 4.69%-9.65% relative error in the artificial dataset, and 0.44%-0.79% relative error in the real-life dataset.