# OpenReview forum: "Branch & Learn for Recursively and Iteratively Solvable Problems in Predict+Optimize"
_NeurIPS.cc/2022/Conference — NeurIPS 2022 Accept_

### Official Review · Reviewer_v3Mn · 2022-07-11

**Rating:** 7
**Confidence:** 3
**Soundness:** 3 good
**Presentation:** 3 good
**Contribution:** 3 good

**Summary:**

This paper proposes the Branch and Learn framework for Predict+Optimize. In this setting, the parameters of an optimization problem are learned from features, and the goal is to predict parameters such that the corresponding predicted solution achieves a low regret under the true parameters.

The proposed framework falls under the category of exact methods, which in contrast to approximate methods do not rely on computing gradients of regret relaxations, at the cost of increased runtime and restriction of the parameter prediction to linear functions.

The framework is applicable to learning the parameters of optimization problems that are solvable via recursive algorithms (under some restrictions). This generalizes previous approaches, which were applicable to iterative and dynamic programming algorithms, to also cover e.g. combinatorial search methods.

On a technical level, the framework follows previous works by performing coordinate descent in the parameter space, i.e. all parameters are held fixed while a single one is updated to achieve the best possible regret over the dataset. To compute this update, it is necessary to construct a function that for a single datapoint computes for every value of the free parameter the corresponding optimal value of the optimization problem. This function is piecewise linear, with the disjoint intervals defined by constant optimal solutions of the optimization problem over the interval. Computing this function is challenging, as depending on the value of the free parameter, the computational flow of the recursive algorithm for solving this problem can dramatically change. The authors provide an algorithm template, which is based on the original recursive algorithm, that computes this piecewise linear function, importantly exploiting the same algorithmic tricks incorporated in the original algorithm (e.g. optimal substructure).

The paper demonstrates the wide applicability of the framework in multiple case studies, covering manifestations of the framework for the shortest path, capacitated min-cost flow, min-cost vertex cover, and the multi-stage scheduling problem.

Finally, the authors conduct an experimental study, comparing their proposed method to multiple classical regression methods as well as three approximation methods and two exact methods. The experiments include the min-cost flow, min-cost vertex cover, and multi-stage scheduling problem, with each experiment considering an artificial dataset and a real-life dataset of varying difficulty and size.

The experiments show that the proposed method consistently outperforms all baselines in terms of mean regret. In terms of runtime, it is comparable to (and mostly faster than) the other examined exact methods, while clearly being much slower than approximate and classical regression methods by design.

**Questions:**

See weaknesses in the quality section: Is there a good reason for the missing comparisons?

**Limitations:**

- The authors have addressed limitations of saclability/runtime.
- The authors have addressed limitations of linear prediction.
- The authors have addressed limitations of requirements on the recursive algorithm, however, it would be good to name an example of an algorithm that does not fit these requirements to give a better idea of the limitations.

**Strengths And Weaknesses:**

### Originality
This paper generalizes and extends previous works. The idea of using coordinate descent with an adapted version of the optimization problem as an exact technique for predict+optimize is not novel, but it is the first method to allow predict+optimize for general recursive algorithms.

### Quality
Overall the quality of the paper is good, the only weakness in terms of quality I could identify was in the experimental comparison to previous works.

I believe a comparison (in terms of runtime and performance on the MCFP and the MSS) to the framework presented in [1], which captures algorithms based on dynamic programming, would have been important, as the presented method claims to generalize this work.

I would also like to point out that a (very similar) version of [2], which is not compared to as it is a very recent publication, already appeared on arxiv [3] in 2020, therefore a comparison to this work could also have been possible, which would have been very interesting given the close relationship between the two methods.
Besides an experimental comparison, it would be good to know if there also exist problems that the framework in [2] can handle, and the presented method cannot.

A minor comment is that it could be good to add some additional more general related work on combining deep learning and constrained optimization, e.g. differentiating through the optimal solutions of convex and combinatorial problems when the optimization problem is not the final component of the architecture.

### Clarity
The clarity of the paper is good.

### Significance
I find the setting of restricting the prediction network to a linear layer very limiting, as it prevents the use of any of the deep learning methods that have proven to be extremely powerful at extracting parameters for decision making even from raw data. Such deep learning approaches are possible with the approximate methods mentioned in the paper, while they are out of reach for the presented method.
However, I agree that there are also important applications of such a shallow learning approach, and in the line of work revolving around this setting, the presented work seems to be significant. It generalizes existing methods, allows the use of previously unusable solver algorithms, and shows superior performance on the conducted experiments.


###### Literature
[1]: E. Demirovíc, P. J. Stuckey, T. Guns, J. Bailey, C. Leckie, K. Ramamohanarao, and J. Chan. Dynamic programming for predict+optimise. In Proceedings of the Thirty-Fourth AAAI Conference on Artificial Intelligence, pages 1444–1451, 2020.

[2]: A. U. Guler, E. Demirovíc, J. Chan, J. Bailey, C. Leckie, and P. J. Stuckey. A divide and conquer algorithm for predict+optimize with non-convex problems. In Proceedings of the Thirty-Sixth AAAI Conference on Artificial Intelligence, 2022.

[3]: A. U. Guler, E. Demirovíc, J. Chan, J. Bailey, C. Leckie, and P. J. Stuckey. Divide and Learn: A Divide and Conquer Approach for Predict+Optimize. https://arxiv.org/abs/2012.02342, 2020.

---

> ### Author Response · Authors · 2022-08-02
> **Authors' response to Reviewer v3Mn**
>
> Thank you for your constructive feedback on improving the paper, and for your positive evaluation of the paper. Here we respond to the points raised in the review.
>
> ---
>
> For the convenience of other readers, please note that the following response uses reference numbers provided by the reviewer and not the reference numbers in our paper.
> + [1] is [5] in the submission.
> + [2] is [9] in the submission.
>
> ---
>
> + There is one minor thing in the summary that we want to point out: the previous approach in [1] can only deal with tabular DP but not iteration in general.
>
> + (Comparing with [1] on MCFP and MSS) We wish to stress again that the framework of [1] cannot be applied to the MCFP and MSS problems. Our framework generalizes that of [1] by solving a significantly larger class of problems than [1] can solve. MCFP and MSS are examples that cannot be formulated as tabular dynamic programming. Therefore, it is impossible to compare with [1] on these problems. Please also see our response to reviewer nihh on comparing [1] with Branch & Learn.
> + (Comparing with the concurrent works [2] and [3], cited as [9] in the submission) Please see our responses to Reviewers 2JwE and nihh.
> + (Compatibility with deep learning; exact methods vs approximate methods in “Significance”) As the reviewer agreed, the uses of deep learning will essentially be restricted to approximate methods and will not be an exact method for differentiability reasons. The main downside of approximate methods is the requirement of using a differentiable surrogate/relaxation of the regret loss, which intrinsically loses information about the problem. It is certainly an interesting and important research direction to investigate whether there exists good surrogates that nonetheless work as well as exact methods (and hopefully even better). However, most importantly, as the experimental results in this paper show, Branch & Learn as an exact method outperforms all the prior approximate methods on our benchmarks, demonstrating the value in pursuing exact methods as a research direction.
> + (Limitations of requirements on the recursive algorithm) Perhaps the most restrictive requirements are the ones on the arithmetic operations we allow in the ReSolve algorithm. One simple (but perhaps unnatural) problem that our framework cannot handle directly is the following: given $n$ items with unknown “rewards” $r_i$, find the two items with the maximum product of their rewards ($r_i r_j$). Computing the product of rewards involves multiplying (instead of adding) unknown parameters, which our framework cannot handle directly. On the other hand, the problem is equivalent to maximizing the sum of the *logarithm* of the two rewards. Thus, in this case, a simple reformulation of the problem makes it amenable to Branch & Learn (though whether the learning would work well is a different question, and depends on the data and the parameters).
>
>     However, we can make the problem and objective slightly more complicated (and more contrived): choose 3 items, and maximize the sum of pairwise products of rewards, as in finding items $i,j,k$ to maximize $r_i r_j + r_jr_k + r_ir_k$. It becomes much less obvious how we can find a *natural* reformulation of the problem that makes it still amenable to Branch & Learn, illustrating some of the limits of the applicability of Branch & Learn.

---

> > ### Comment · Reviewer_v3Mn · 2022-08-08
> > **Answer to rebuttal**
> >
> > I want to thank the authors for addressing my concerns in their rebuttal. My main concerns were on missing experimental comparisons. I now agree that I misunderstood the relationship to [1], which makes a comparison on the suggested experiments obsolete. Regarding the relationship to [2] and [3], I like the clarifications made in the authors answer to the other reviewers, and I would like to see parts of these clarifications in a revised version of the paper (as the original version of the paper apparently left multiple reviewers with questions regarding this relationship).
> > The rebuttal also gives a convincing justification for using exact over approximate methods, even though the machine learning side of it is limited to linear predictions.
> > Finally, the additional limitation discussed by the authors arising from restrictions on the arithmetic operations is quite interesting. I believe this discussion would also be useful to a practitioner who wants to employ the proposed framework, therefore I would suggest including parts of this discussion of limitations in the paper.
> > Overall, the answers of the authors to the raised concerns were convincing. Therefore I will raise my rating once a revised version of the paper incorporating the small modifications is available.

---

> > > ### Author Response · Authors · 2022-08-09
> > > **Revision uploaded**
> > >
> > > We thank the reviewer for further comments, and are encouraged by the positive response. We have revised the paper (uploaded on OpenReview) as follows, highlighted in blue in the pdf:
> > >
> > > + At the end of Section 1, we mentioned that [3] and our work do not subsume each other, and added a section (Section 6 in the revised version) between “Case Studies” and “Experimental Evaluation” for a more detailed comparison of Branch & Learn and the work of Guler et al.  The paper is 10 pages now, but we are allowed an additional page if the paper is accepted.
> > >
> > > + At the end of Section 3, we added a paragraph to show examples of the limitations of the requirement of the ReSolve template.
> > >
> > > We definitely welcome any additional suggestions to improve the paper.

---

### Official Review · Reviewer_nihh · 2022-07-11

**Rating:** 6
**Confidence:** 3
**Soundness:** 3 good
**Presentation:** 3 good
**Contribution:** 2 fair

**Summary:**

In the Predict+Optimize framework, parameters (costs) of an optimization problem are predicted from features and then processed by a combinatorial algorithm (solver). This paper fits a narrower line of P+O works that aims to learn a linear model minimizing the regret (difference of predicted and true objective value). As the regret is non-differentiable, standard gradient-based methods cannot be used. Existing methods either use surrogate gradients (called approximation methods, like SPO [6] for instance) or coordinate descent with exhaustive search (called exact methods). Exact methods are less general (linear model), require more resources (large run times), but achieve better performance (smaller regret).

This paper builds on an exact method of Demirovic et al. [5] that allows for using any tabular dynamic programming algorithm (DP) as a solver and extends it for optimization problems solvable by a recursive algorithm (under some mild conditions). The authors explain the background in detail, describe their approach (denoted as B&L) and how it covers existing algorithms. Finally, they experimentally show the method's performance on existing and also one new case-studies compared to competing methods.

**Questions:**

- I did not understand the run times scaling paragraph. From Algorithm 1 it seems that the method scales linearly with dataset size $n$ which is also in consent with the experimental results. Can you also comment why SPO tree and SPO forest do not scale linearly as well?
- Coordinate descent may get stuck at a non-stationary point if the level sets are not smooth. In theory, this may occur. Does it happen in practice? Can the coordinate descent be adjusted to other than canonical basis?
- Why don't you include the experimental comparison to [5] when applicable, as B&L generalizes it? The performance should be the same, but the runtime comparison might be useful.
- You claimed that method of [9] is not applicable to Multi-stage scheduling algorithm. Does B&L also generalize this work, or is there any real-world applicable algorithm that fits [9] and not B&L?

**Suggestions**
- Algorithm 2, Function ReSolve: $R$ is not defined, should be $(x^*,C^*)$ instead?
- The tables with experiment results are hard to parse. A (box)plots would be much more reader-friendly.

**Limitations:**

The authors addressed the limitations and I found no negative societal impact of their work.

**Strengths And Weaknesses:**

- The method is new and generalizes [5] from DP to recursive algorithms.
- The article is logically organized, well written, typed with care without many misprints. It is easy to read; I found it as one of the most comprehensible in this line of work.
- My major concern is in significance. This research area is very narrow and fits more into operations research community, where the domain of use is fixed and the performance requirements outweight the large runtimes. I do not think that it is very likely that NeurIPS audience would benefit much from or build upon ideas presented in this paper.

---

> ### Author Response · Authors · 2022-08-02
> **Authors' response to Reviewer nihh (1/2)**
>
> Thank you for your constructive feedback on improving the paper, and for your overall positive assessment of our paper.
>
> Before addressing your questions, we would like to respond to your concern about the fit of our paper in NeurIPS. While Predict+Optimize does have some operations research flavor, it is nonetheless fundamentally a machine learning problem. We also note that the NeurIPS+ICML community has shown interest in this line of work. This is evidenced by two prior works in Predict+Optimize that have been accepted in past ICML or NeurIPS ([7] in ICML 2020, an exact method, and [12] in NeurIPS 2020, an approximate method). Therefore, we decided to submit this work to NeurIPS.
>
> We now answer the questions raised in the review.
>
> 1. (Runtime scaling for SPO Tree/Forest) For space reasons, we only presented two points in the scaling of runtime in the dataset size, which makes it difficult to infer runtime scaling. Here we give a very high-level sketch of what the runtime of SPO Tree scales like.
>
>     Having investigated the actual implementation for SPO Tree, its runtime can be roughly modelled by the function $\beta \cdot \min(m, 100) + O(m)$ where $\beta$ is the (expensive) runtime of solving an optimization problem, and $m$ is a quantity upper bounded linearly by the dataset size $O(n)$. Essentially, $m$ is related to the number of distinct feature values in the training data. The min with 100 in the expression is due to the algorithm only examining the 100 percentiles of lists of distinct values. In other words, the runtime is in fact actually bounded linearly in the datasize $n$, although there is a very high cost associated with solving the optimization problem repeatedly. In our experiments, when the dataset size $n$ is 100, the quantity $m$ is somewhat small, and when $n = 300$, $m$ gets close to 100. It is for this reason that we see a (very) large jump in the runtime.
>
> ---
>
> 2. (Coordinate Descent: non-stationary points and change of basis) We did not investigate whether coordinate descent gets stuck at a non-stationary point in our experiments, but we did check that they at least converge to some point (ruling out the even-worse scenario where the level set of the regret is some axis-aligned cuboid, and the coordinate-descent just keeps bouncing around). Note that, even if our method is potentially getting stuck at non-stationary points, it is nonetheless outperforming all the other methods in terms of the generalization error in the regret.
>
>     As for doing coordinate-descent on a non-canonical basis, we had not considered this and it is a nice idea. It is possible with our framework. Even after a change of basis, the parameters can still be expressed as a linear transformation of the new coefficients. The rest of the B&L framework would go through in exactly the same way.

---

> > ### Author Response · Authors · 2022-08-02
> > **Authors' response to Reviewer nihh (2/2)**
> >
> > 3. (Runtime comparison with DP method in [5]) All benchmarks in our paper cannot be handled by the method of [5] since we aim to demonstrate how Branch & Learn can tackle interesting problems that are otherwise not possible with [5].
> >
> >     As you have noted, for problems that have a tabular DP algorithm, our method would have identical performance as [5], and it would be interesting to compare the runtimes. We did not present these results partly for space reasons, and partly because the runtime comparison might depend more on the software engineering than on the intrinsic overhead (or lack thereof) of our approach. For space, we prioritized demonstrating problems that [5] (and other exact methods) cannot handle.
> >
> >     When given a tabular DP algorithm, our recommendation of course is to directly use the approach and implementation of [5], since they have a specialized implementation tailored to the simplicity of tabular DP algorithms.
> >
> >
> > 4. (Comparing with [9]) Branch & Learn and [9] do not subsume one another. There are problems amenable to Branch & Learn but not [9], and vice versa. Here we can give some intuition about when each method is applicable.
> >
> >     + Suppose we have an optimization problem with a finite feasible solution set. For most practical purposes, such a solution set is enumerable by a recursive algorithm (but not always, since finite sets need not be enumerable for computability reasons). If, additionally, the objective function is bilinear as in the assumption of [9], then Branch & Learn would be able to handle such a problem. In this sense, an optimization problem with a finite solution set that is solvable by [9] would also be solvable by Branch & Learn.
> >
> >     + On the other hand, if the solution space is continuous, Branch & Learn might not be able to handle such an optimization problem. The reason is that Branch & Learn (as it is currently formulated) requires explicitly computing the entire empirical regret function, which is possible only if the regret has a finite representation (say, as a piecewise linear function). When the solution space is continuous, the regret might be a general smooth function even if the objective is bilinear. By contrast, [9] never computes the entire regret function explicitly, and instead just queries the regret function and computes its values, then performs numerical optimization on the function (which is convex when the objective is bilinear). In this sense, [9] is more capable than Branch & Learn in these continuous settings.

---

### Official Review · Reviewer_1y7n · 2022-07-11

**Rating:** 5
**Confidence:** 3
**Soundness:** 3 good
**Presentation:** 1 poor
**Contribution:** 2 fair

**Summary:**

The paper considers the predict+optimize problem where a constrained optimisation problem contains parameters that are unknown at the time of solving. Specifically, a branch&learn framework is developed to deal with problems solvable by a recursive or an iterative algorithm. The framework extends recent work that focuses on dynamic programming type of algorithms. The experimental evaluation is carried out on an extensive set of benchmarks and demonstrates the superiority of the proposed approach compared with existing state-of-the-art methods.


**Questions:**

1. In the experimental evaluation, how many intervals do you consider for the piecewise linear functions? How do you choose the intervals?

2. Dynamic programming can also be viewed as a recursive procedure but in the opposite direction and therefore I'm wondering if the approach described in Demirovic et al AAAI-20 paper is equivalent to the one proposed here. Is recursion here meant to capture search-based algorithms like depth-first branch and bound?

[Post Rebuttal] Thanks for your answers. They clarified some of my concerns.

**Limitations:**

see above

**Strengths And Weaknesses:**

The paper is fairly well written and organised. The experimental evaluation appears to be comprehensive in terms of benchmark problems and includes relevant baselines.

However, a major weakness of the paper is the quality of the presentation which is rather poor in my opinion. I think the paper needs a more detailed running example that would add more clarity to the technical contribution. Examples 1, 2, and 3 are good but they are too high level to help the reader get a better understanding of the proposed algorithms. I suggest adding more details to the project funding example such as: a possible choice of intervals for the piecewise linear function, the search tree that illustrates the recursion process, as well as example features and training data that is used to learn the unknown parameters.

---

> ### Author Response · Authors · 2022-08-02
> **Authors' response to Reviewer 1y7n**
>
> Thank you for your feedback on the paper. We agree that the paper would benefit from more detailed examples, but we are constrained by the page limit. If accepted, we will try our best to include more details in the final version with the allowed extra page.
>
> We now address the concrete questions raised in the review.
>
> 1. An important point to emphasize here is that the intervals in the piecewise linear empirical regret function are inherent in the function itself (depending on the problem structure, as well as the training data). Our method *automatically* finds all the interval boundaries. These intervals are not hand-picked or hardcoded in any way. $\qquad\qquad\qquad\qquad\qquad\qquad\qquad\qquad\qquad\qquad\qquad$
>
>     As for the number of intervals, since the empirical regret function is inherent in the problem and depends on the training data, the number of intervals in it can vary, ranging from constant to linear to exponential in the relevant problem parameters.
>
>     + The constant regime happens for example with optimization problems with no unknown parameters.
>
>     + For an example of the linear regime, consider the simple problem of having $n$ items with unknown rewards, and the goal is to choose the one with the largest reward. Assuming all the features are positive, then the predicted optimal value as a function of a regression coefficient is the max of $n$ linear functions (the reward of each item is a linear function $a\gamma + b$ for some values $a$ and $b$, for the regression coefficient $\gamma$. See lines 100-103 in the paper for details). As such, the predicted optimal value has only $n$ intervals. The regret is the true optimal value (a constant) minus the predicted optimal value, and hence it also has at most $n$ intervals.
>
>     + Our benchmark, the minimum cost vertex cover problem, is an example of the exponential regime, where the number of intervals is exponential in the number of vertices. This can be demonstrated and checked empirically.
>
> ---
>
> 2. Recursion in general cannot be expressed as tabular dynamic programming (at least, not without unreasonable twisting. See also lines 38-43). Indeed, the recursion our work can handle includes (but is not limited to) depth-first tree search, and our case study on minimum cost vertex cover is an example of that. And of course, all tabular DP algorithms can be implemented as an iterative algorithm (which is a degenerate form of recursion). Therefore, Branch & Learn applies to all problems solvable by [5].

---

### Official Review · Reviewer_2JwE · 2022-07-12

**Rating:** 6
**Confidence:** 1
**Soundness:** 3 good
**Presentation:** 3 good
**Contribution:** 3 good

**Summary:**

The paper presents a novel predict+optimize framework, named Branch and Learn, that extends the work of [5] to problems solvable via a recursive algorithm. Experimental results show that Branch and Learn performs better than competing methods and supports a wider variety of benchmarks.

**Questions:**

Could the authors detail the high-level differences to [9]?

**Limitations:**

It would be nice if the authors could group all the limitations of their work in a section in the appendix.

**Strengths And Weaknesses:**

The experimental results consistently show that Branch and Learn yields lower regret than many existing methods on a variety of benchmarks, without incurring significant overheads.

---

> ### Author Response · Authors · 2022-08-02
> **Authors' response to Reviewer 2JwE**
>
> Thank you for your positive appreciation of our work.  We explain the high-level differences between our work and [9] as follows.
>
> + Both Branch & Learn and [9] are exact methods for Predict+Optimize, since they both directly optimize the regret, instead of any surrogates. Both also use coordinate-descent as the high-level optimization routine. The main differences lie in the kind of optimization problems that the methods can apply to, as well as the (very different) techniques used to achieve each coordinate-descent step.
>
> + In terms of the applicable problems, Branch & Learn applies to all optimization problems amenable to an optimization algorithm fitting the ReSolve template. On the other hand, [9] applies to optimization problems where the objective function is bilinear in both the decision variables as well as the unknown parameters, i.e., the objective is the inner product between the two vectors. As we demonstrated in the paper (see also lines 282-284 in the paper), the multi-stage scheduling problem with unknown per-item/machine processing times is an example problem with no such bilinear-objective representation (unless we use an exponentially large integer linear program to represent the problem, with exponentially many unknown parameters), yet is easy to fit into our framework. For a discussion on whether Branch & Learn generalizes [9] (which is rather subtle), please refer to our reply to Question 4 for Reviewer nihh.
>
> + In terms of techniques, Branch & Learn automatically generates the “transition points” in the piecewise linear functions by essentially simulating the recursive algorithm. This contrasts the approach of [9], where they leverage convexity structure in the problem arising from their bilinearity assumption on the objective, and they use a numerical optimization approach (essentially a variant of binary/ternary search) to perform a coordinate-descent step.

---

> > ### Comment · Reviewer_2JwE · 2022-08-08
> > **Thank you for your response.**
> >
> > Thank you for your response. I think the paper would greatly benefit from the inclusion of the above discussion.

---

> > > ### Author Response · Authors · 2022-08-09
> > > **Revision uploaded**
> > >
> > > Thank you for following up. We have uploaded a revision of the paper, with changes highlighted in blue. The new Section 6 includes a more detailed discussion comparing our work and the work of Guler et al.

---

### Author Response · Authors · 2022-08-02
**Overall response**

Thank you for your constructive and detailed feedback for improving the paper. We will address your remarks and questions separately as responses to each individual review. Here, we wish to emphasize that:

+ Our work significantly generalizes the work of [5].  Tabular DP can be implemented as an iteration (a degenerate form of recursion), and recursion in general *cannot* be (naturally) expressed as tabular DP.

+ The work of [5] is a special case of our Branch & Learn framework.  Thus, if we apply [5] and Branch & Learn on problems that both can solve, the results will be identical in terms of regret quality.  Our case-study+benchmark problems of MCFP, MCVC and MSS are examples that cannot be handled by the framework of [5]. It is therefore impossible to do a comparison with [5] on these problems.

---

### Meta-Review · Area_Chair_4HNC · 2022-08-24

**Recommendation:** Accept
**Confidence:** Certain

**Metareview:**

This paper considers the general setting of the predict+optimize framework, where "optimize" part of the problem is typically solved via a recursive algorithm. The paper proposes a new exact algorithm to directly optimize the regret in this setting. An extensive evaluation of the new methodology is also provided.

This paper represents a significant generalization of the existing techniques and is definitely of interest to Neurips community.

**Award:**

No

---

### Decision · Program_Chairs · 2022-09-14

Accept